# Destigmatizing Palliative Care among Young Adults—A Theoretical Intervention Mapping Approach

**DOI:** 10.3390/healthcare12181863

**Published:** 2024-09-16

**Authors:** Yann-Nicolas Batzler, Manuela Schallenburger, Jacqueline Schwartz, Chantal Marazia, Martin Neukirchen

**Affiliations:** 1Interdisciplinary Centre for Palliative Medicine, Medical Faculty and University Hospital Düsseldorf, Heinrich-Heine-University Düsseldorf, 40225 Düsseldorf, Germany; yann-nicolas.batzler@med.uni-duesseldorf.de (Y.-N.B.); jacqueline.schwartz@med.uni-duesseldorf.de (J.S.); martin.neukirchen@med.uni-duesseldorf.de (M.N.); 2Department of the History, Philosophy and Ethics of Medicine, Centre for Health and Society, Medical Faculty, Heinrich-Heine-University Düsseldorf, 40225 Düsseldorf, Germany; chantal.marazia@med.uni-duesseldorf.de; 3Department of Anesthesiology, Medical Faculty and University Hospital Düsseldorf, Heinrich-Heine-University Düsseldorf, 40225 Düsseldorf, Germany

**Keywords:** palliative care, intervention mapping, young adults, stigma, public health

## Abstract

*Background:* In medicine, stigmatization pertains to both afflicted individuals and diseases themselves but can also encompass entire medical fields. In regard to demographic change and the rising prevalence of oncological diseases, palliative care will become increasingly important. However, palliative care faces multiple stigmas. These include equating of palliative care with death and dying. A timely integration of palliative care would have the potential to alleviate symptom burden, diminish the risk of overtreatment, and thus save healthcare-related costs. Several interventions have been developed to destigmatize palliative care. However, they have mainly focused on the general public. *Aim:* The aim of this work is to develop a theoretical framework for an interventional campaign targeted at young adults to systematically destigmatize palliative care. *Methods:* The basis for the development of the campaign is a systematic review conducted by our working group that assessed the perception and knowledge of palliative care of young adults aged 18 to 24 years. To design a possible intervention, the Intervention Mapping approach was used. *Results:* The target group of young adults can be effectively reached in secondary schools, vocational schools, and universities. The target population should be able to discuss the content of palliative care and openly talk about death and dying. At the environmental level, palliative care should be more present in public spaces, and death and dying should be freed from taboos. Within an intervention with palliative care experts and patients serving as interventionists, these changes can be achieved by incorporating evidence-based methods of behavioral change. *Conclusions:* An early engagement with palliative care could contribute to the long-term reduction of stigmas and address the demographic shift effectively. A multimodal intervention approach comprising knowledge dissemination, exchange, and media presence provides an appropriate framework to counter the existing stigmatization of palliative care within the peer group of young adults.

## 1. Introduction

In medicine, stigmatization pertains to both afflicted individuals and diseases themselves but can also encompass entire medical fields, thus being of high relevance in public health [1]. Stigma is considered a fundamental cause of health inequalities and a “central driver of morbidity and mortality at a populational level” [2]; as such, it is a relevant issue in public health. Individuals affected by stigmatization due to illness may experience self-stigmatization as societal indoctrinated beliefs are internalized [3]. Values and beliefs learned in life as a healthy individual are transferred into situations of illness. This can lead to shame, fear, and diminished self-esteem. Social exclusion resulting from stigmatization has implications for both mental and physical well-being, thereby exacerbating the burden of disease. As a result of experienced illness-related stigma, healthcare services are utilized to a lesser extent [1,4,5]. Treatment initiation is delayed or initiated belatedly. Furthermore, due to feelings of shame, stigmatized patients exhibit poorer compliance and therapy adherence [1]. Additionally, health professionals may not entirely disassociate themselves from stigmatization processes, leading to compromised treatment quality [6]. Consequently, morbidity and mortality rates increase, further exacerbating social inequality [6].

In regard to demographic change and the rising prevalence of oncological diseases, a medical discipline that will become increasingly important in the future for many affected individuals is palliative care [7]. However, this relatively young medical discipline itself faces multiple stigmas.

### 1.1. Stigmatization and Palliative Care

Palliative care is associated with death and dying [7,8], and is perceived to primarily focus on end-of-life treatment for older people that suffer from cancer [9,10,11]. According to Link and Phelan’s model [12], stigmatization is defined by the (joint) co-occurrence of the following phenomena: labeling, stereotyping, loss of status, and discrimination [6,13] (Figure 1). When transposed onto palliative care, the following causal interacting mechanisms can be elaborated. Due to the COVID pandemic and a rising prevalence of international conflicts, diseases and death are more present in the mass media and, as a result, in people’s lives. However, in personal spaces, death and dying are still mainly avoided in everyday life. Severely ill individuals deviate from the norm of health and are downgraded to their illness (labeling). Historically, these individuals were cared for in hospices [14,15], more recently also by palliative care, leading to a potential stigma associated with palliative care: palliative care solely addresses the treatment of terminally ill individuals at the end of life (stereotyping). The exclusion of palliative care from daily life leads to an exclusion of death and dying. Structural discrimination ensues, which in turn leads to patients who would benefit from the integration of palliative care refraining from seeking it due to fear of social exclusion [16].

A timely integration of palliative care within disease trajectories would have the potential to alleviate symptom burden, diminish the risk of overtreatment at the end of life, and thus save healthcare-related costs [17]. Due to stigma, these aspects are omitted by both patients and healthcare personnel, thus leading to poorer quality of care [18]. Consequently, resource allocation is poor, which makes the stigma surrounding palliative care a major public health-related problem. In order to promote a timely integration of palliative care, as demanded by international guidelines and the World Health Organization, stigmas have to be diminished [8,19].

### 1.2. Reduction of Stigmas

In the context of public health, the focus should be on reducing stigma through targeted interventions and public campaigns [4,5,20,21,22,23]. Designing such interventions, especially in the context of palliative care, however, is difficult [24,25,26]. Studies on destigmatization should incorporate cultural aspects, and changes over time should be documented [21]. Marginalized groups experiencing social exclusion should be integrated into research endeavors on destigmatization. However, recruiting these groups is challenging as evaluation results of interventions may potentially further marginalize them [27]. In addition to quantitative methods, particular attention should be paid to qualitative studies, especially in the realm of public destigmatization efforts. These studies are participatory in nature and aim to involve stakeholders. This leads to empowerment for the affected individuals, enabling them to shape their health behaviors [17,21,28]. Various intervention forms are described for such efforts. These include protest, contact, and educational strategies [5]. In the protest strategy, there is direct confrontation with stigmatizing individuals or groups. Contact strategies involve stigmatizers directly interacting with stigmatized individuals to raise awareness of stigmas and their consequences through interaction. Education strategies involve imparting knowledge on specific topics [5]. Interventions should always draw upon several of these strategies to target as many mechanisms of stigmatization as possible [5].

### 1.3. Intervention Planning

In the context of public health, prevention and health promotion play a crucial role. To counteract risk behaviors at the population level, structured and well-implemented interventions are necessary. A theoretical foundation for this is provided by the “Public Health Action Cycle”, which defines crucial steps (assessment, policy formulation, assurance, evaluation) in the development of interventions [29]. The PRECEDE/PROCEED model, introduced by Green and Kreuter in 2005, views problems from a socio-ecological perspective, incorporating considerations based on societal and environmental factors [30]. The model is divided into two phases. In the PRECEDE phase (an acronym for Predisposing, Reinforcing, and Enabling Causes in Educational Diagnosis and Evaluation), the target population, along with their living conditions and factors affecting their quality of life, are defined. In the PROCEED phase (an acronym for Policy, Regulatory, and Organizational Constructs in Educational and Environmental Development), the implementation of the intervention takes place. This phase is followed by an evaluation of the intervention, the behavioral changes achieved, and the impact on the health and quality of life of the target population. Similar to the PRECEDE/PROCEED model, the “Intervention Mapping” model, developed by Bartholomew Eldredge and colleagues, views health-related problems as a multifactorial interplay of social and environmental factors [31].

### 1.4. Intervention Mapping

The concept of Intervention Mapping is well-suited for interventions aimed at destigmatization thanks to the provision of a clearly formulated structure. Often, it is challenging to disseminate intervention programs beyond the pilot phase. This is partly due to the difficulty of gaining access to and engaging with stigmatized target groups [32]. Additionally, concrete implementation plans are often lacking, which is essential for the success of any intervention. Intervention Mapping provides a definite framework from conception through dissemination to evaluation. It thrives on a participatory character, where affected individuals and those who stigmatize work together. Intervention Mapping is not a strictly linear approach but rather thrives on flexibility, allowing for modifications and adaptations of initial steps during the design process. Given that stigma is a procedural issue upheld by socio-cultural constructions, a flexible approach is appropriate [32]. Besides these participatory and socio-ecological approaches, Intervention Mapping employs a problem-based approach, addressing behavioral change starting from a specific problem and pursuing it through multiple layers. This aligns with the nature of stigmatization itself, which, as previously described, is not singular but multifactorial and multidimensional [28,33]. At our faculty, this methodology is regularly used in the development of health-related interventions such as encouraging the general public at increasing daily steps. Currently, it serves as the basis for the development of a palliative care awareness month in Düsseldorf, Germany (work in progress).

### 1.5. International Campaigns

Internationally, several high-profile campaigns and interventions have been developed to destigmatize palliative care.

In the United States, the “Project on Death in America” aimed to reduce stigma surrounding palliative care, death, and dying through various initiatives [34]. As part of this project, a radio series addressing end-of-life issues was produced. Additionally, an exhibition at the Strong Museum in Rochester, New York, displayed images and artifacts expressing grief. Information on accessing end-of-life medical services was disseminated, and myths about death and dying were dispelled through knowledge dissemination in a national magazine called “Aging Today” (1996). In Ireland, an annual “Palliative Care Week” aims at raising awareness [35]. Informational brochures are produced, and efforts are made to increase the media presence of palliative care. Individuals affected by serious illnesses share their stories on radio and television, while healthcare professionals provide information on both inpatient and outpatient options in palliative care. In England, the “Dying Matters” campaign seeks to encourage the general population to discuss end-of-life wishes and foster an open dialogue about death and dying [36]. In Canada, the “Canadian Virtual Hospice” website serves as a virtual hub for information on palliative and hospice care [35]. A campaign was launched to debunk common myths about palliative care through empirical evidence. In Germany, a project targeted towards children in elementary schools is called “Hospiz macht Schule” (English: hospice makes school) [37]. During five days, specially trained teams teach children about death and dying and interact with them within small groups. Furthermore, the German Society for Palliative Medicine launched the campaign “das ist palliativ” (“this is palliative”) at the end of 2022 [38]. The campaign aims at educating the public about the components of palliative care, encouraging active engagement with one’s own mortality, and promoting advance care planning for end-of-life care. Prominent persons have been enlisted to convey key messages about palliative care through concise, impactful statements.

However, all of these programs have mainly focused on the general public and none of them focused especially on young adults. This age group, however, is of great relevance in the context of public health, as it is during this period in life that the foundations for health awareness are established. Furthermore, health behaviors can still be influenced [39,40], and dealing with health-related issues is ingrained in everyday life through familial experiences and peer interactions [40].

### 1.6. Aim

It is the aim of this work to develop a theoretical framework for an interventional campaign targeted at young adults to systematically destigmatize palliative care following the Intervention Mapping approach.

## 2. Material and Methods

The basis for the development of the campaign is a systematic review conducted by our working group that assessed the perception and knowledge of palliative care of young adults aged 18 to 24 years [41].

The German Federal Ministry of Justice and Consumer Protection, for instance, defines individuals aged 18 to 27 as “young people” [42]. Internationally, there is no uniform definition, so for this study, the age range was adjusted: individuals aged 18 to 24 years were defined as the target population as they can be systematically reached in educational and vocational institutions.

On the basis of the PRISMA Checklist and according to the PICOS process, a search string was developed. Within Medline (via Pubmed), Google Scholar, and Web of Science, relevant literature addressing young adults’ knowledge and perception of palliative care was screened by two reviewers.

To design a possible intervention, the Intervention Mapping approach was used. This model employs a participatory approach, involving representatives of the target population, stakeholders, and experts in planning the intervention [31,32].

This is in accordance with Kellehear’s ‘new public health’ approach to palliative care, as affected individuals seek support from peers, family, and friends in severe disease trajectories [43]. To empower them and further develop crucial networks, so called compassionate communities are developed. It is them who need to play an active role in the development of a novel intervention.

The Intervention Mapping model comprises six steps that can follow each other but are not to be seen as a rigid structure; returning to a previous step and adapting results based on new insights is possible [32]. The six steps of Intervention Mapping are as follows:Logic Model of the Problem: Needs AssessmentLogical Model of Change: Defining Intervention Goals and Determinants of Behavioral ChangeDesign of the Intervention through the Targeted Selection of Theory- and Evidence-Based Methods for behavioral ChangeProgram Organization and Design, including PretestingImplementationEvaluation Planning.

In the first step, a planning group is formed and a needs assessment is conducted to create a logic model of the problem. Furthermore, the population and setting are defined and initial goals of the intervention are formulated. In the second step, expected goals for behavioral change are defined, and so-called “Performance Objectives” are developed. Important modifiable determinants are listed in this step, creating a logic model of behavioral change. In the third step of Intervention Mapping, the themes and content of the intervention are determined, and the sequence of individual components is established. Theory- and evidence-based methods are selected to achieve behavioral change, and their practical implementation is planned. In the fourth step, the structure of the intervention is refined, and program materials are produced. These contents are pretested and potentially further refined. In the fifth step, the key actors implementing the intervention are identified. To ensure the success of the intervention, key actors should already be included in the planning of the intervention’s execution. In the final step, an evaluation plan is developed [32].

Since this study was exclusively based on literature and theoretical developments, an ethical committee approval was not required.

## 3. Results

To effectively engage young adults, this target group should be included in the planning, feasibility assessment, and evaluation of content. As learned from the above-mentioned systematic review, new media platforms are preferred for disseminating information and influencing perceptions of palliative care within this demographic.

The findings from the systematic review [41] provide a solid foundation for developing ideas for public health interventions and awareness campaigns aimed at destigmatizing palliative care among young adults. Using the Intervention Mapping model, these ideas can be conceptualized within a theoretical framework for the various components of such an intervention.

### 3.1. Intervention Mapping

For the target population of young adults, the following ideas for potential components of an intervention aiming at destigmatizing palliative care were developed by the working group based on the findings of the systematic review [41]:Theoretical knowledge dissemination in secondary schools, vocational schools, and universities;Interactions with affected individuals and palliative care teams;Public awareness campaigns in public spaces;Public awareness campaigns on social media;An accompanying website with a discussion forum.Their concrete elaboration follows the Intervention Mapping approach.

#### 3.1.1. Step 1: Needs Assessment and Logic Model of the Problem

The central foci of the first step of Intervention Mapping are the needs assessment and problem definition. Additionally, a planning group should be established. Young people desire participation in the planning and design of interventions and public awareness campaigns. They should be included as representatives of their peer group in this planning group. To capture a wide range of social backgrounds, multiple representatives of different ages and from various educational institutions (middle school, university, vocational school) should be involved. Additionally, representatives from the Ministry of Education, school principals, and university representatives should be included. To disseminate information, social media experts and representatives from the Ministry of Health should be present. Public health experts will complete the planning group.

The needs assessment and problem definition were conducted using the systematic review findings. They serve as the basis for the development of the logic model of the problem. In regard to knowledge and perception, the following key aspects should be targeted (see Figure 2):-Regarding knowledge of palliative care, the following were found:Young adults often have heard of palliative care;Young adults, however, have little detailed knowledge of palliative care;Young adults assume that palliative care is only relevant in the last six months of a person’s life.-This affects the perception and attitude towards palliative care:Palliative care is seen as indicated only for certain illnesses;Palliative care is closely associated with death and dying, which can trigger fears;Death and dying are associated with hopelessness, and thus palliative care is also linked with giving up and hopelessness.

On the other hand, young adults demand that palliative care is destigmatized and normalized within the education system. They particularly advocate for participation in the development of educational content. The analyzed studies repeatedly emphasized a participatory approach in the development of public health campaigns aimed at destigmatization of palliative care.

Based on these findings, the following program goals were defined:-Increase in detailed knowledge of palliative care: palliative care is available for patients of any age at any stage of illness and follows a holistic approach.-Change of the perception of palliative care: palliative care is life-affirming, improves quality of life, and conveys hope.

The target group of young adults can be effectively reached in social contexts such as secondary schools, vocational schools, and universities. Attitudes and values are malleable and influenceable, and information dissemination typically occurs through exchanges within peer groups [39,40]. The systematic review highlighted that the internet and social media are preferred resources for acquiring knowledge. Social media, in particular, offers quick access to information and provides a platform for discussing difficult topics in an anonymous and private setting. Therefore, establishing an accompanying website is advisable. This site should be optimized for smartphone use, as familiar devices are preferred. To counter the spread of misinformation, health experts should monitor comments and participate in forums to interact with users.

Since stigmatization of palliative care is not limited to the peer group of young adults, the taboo around discussions of palliative care is perpetuated and an open societal discourse is prevented. Therefore, it is not enough to only consider the personal determinants of the target population; behavioral change must also be achieved at the environmental level, e.g., through environmental agents such as healthcare personnel. An intervention to destigmatize palliative care should thus occur in an intrapersonal, interpersonal, and institutional context.

These environmental and personal determinants lead to the development of the logic model of the problem for this study’s purpose (see Figure 3):

#### 3.1.2. Step 2: Logic Model of Change

In the second step of Intervention Mapping, the goals of the intervention at the behavioral and environmental levels are more precisely defined. Action goals, known as “performance objectives”, and determinants for the behavioral and environmental goals are identified, upon which the “matrices of change” are based.

At the behavioral level, the target population should be able to discuss the content of palliative care and openly talk about death and dying. At the environmental level, palliative care should be more present in public spaces and death and dying should be freed from taboos. This can be achieved through public awareness campaigns using posters or key messages on social media.

The performance objectives describe in detail how the target population and environmental actors should act to achieve the program goals:-Young adults:… discuss acquired knowledge of palliative care with each other;… talk about death, dying, and dealing with diseases to each other.-Environmental actors:Educational institutions facilitate education on palliative care;Educational institutions distribute information on palliative care;Communities place posters in public spaces;Information on palliative care is openly accessible via the internet.

In the domain of personal determinants, attitude, knowledge, and values are crucial and must be altered through the intervention. Table 1 summarizes the performance objectives on the level of personal determinants.

In the area of environmental determinants, knowledge and attitude are equally important as on the personal level. They further concretize performance objectives on the environmental domain (see Table 2).

#### 3.1.3. Step 3: Components and Behavior Change Methods

In the third step of intervention mapping, the components of the intervention are generated. To precisely define change processes, theory- and evidence-based change methods are selected based on the previously identified determinants.

A campaign to destigmatize palliative care should incorporate the following components:-Intervention components:Knowledge dissemination about palliative care in educational institutions;Interaction with palliative care patients and staff;Informational brochures.-Public awareness campaign:Website;Social media presence;Posters in public spaces.

In order to achieve the goal of a profound behavioral change, theoretical and evidence-based change methods that influence both knowledge and behavior change are selected [31].

Methods to increase knowledge about palliative care include “discussion” and “chunking”. For instance, within educational institutions, a video depicting typical routines in a palliative care unit could be shown, followed by a discussion among intervention recipients about the content. In the context of chunking, acronyms associated with learned material can be used to reinforce knowledge. One example could be the word “palliative”, which could be associated with positively connoted terms.

“Classical conditioning”, where positive attributes are conveyed through moving images during knowledge transfer, serves as a tool for behavior change. This could involve depicting severely ill individuals navigating their daily lives positively without symptomatic burdens.

To reduce stigma, further behavioral methods are found in the literature. In the context of destigmatizing palliative care, “interpersonal contact” seems essential, where participants in the intervention could be brought into direct contact with palliative care patients to reduce prejudices. Patients with stable diseases and overall well-being could be present at educational institutions during the course of the intervention. They can be recruited through palliative care teams serving as advisors.

“Empathy training” is also crucial here, as people recount experiences to build empathy. In palliative care, this could involve patients sharing experiences of feeling excluded due to their illness. “Cooperative learning”, in which information is distributed to individual students who then work in groups to piece it together, is also suitable for reinforcing knowledge and learning from one another.

In the area of normative determinants, “role modeling” is considered important. Affected individuals could report that early integration of palliative care led to significantly better tolerance of systemic therapies and improved quality of life and helped in prolonging life. Furthermore, “mobilizing social networks” is vital, as it addresses the surrounding environment. Here, information is disseminated within the target population’s environment to pass on and reinforce knowledge. “Community development” plays a crucial role in disseminating palliative care content: communities are actively involved in the intervention as stakeholders discuss problems and develop solutions together.

#### 3.1.4. Step 4: Program Organization and Design

In the fourth step of intervention mapping, the contents and components of the intervention are refined. Additionally, materials to be used in the intervention are created. These should be relevant and help achieve the intervention’s goals, serving as supportive rather than core elements of the intervention. The systematic review indicated that younger people, men, and ethnic minorities are less informed about palliative care. This could be addressed in the developed intervention. For example, a video sequence featuring a young male patient with a specific cultural background sharing positive experiences with palliative care could be produced. Alternatively, in-depth informational brochures could be utilized. A website for the intervention should also be launched, accessible via QR code, which provides additional materials like videos. It is crucial to conduct a “reality check” to ensure program materials reach the participants, with target population representatives playing a key role in this process. Budget and timeline should also be reviewed. Pretesting and pilot testing are conducted in this step. Pretesting checks if individual components and materials achieve the desired effect. Pilot testing involves implementing the entire intervention on a small scale before full implementation, such as within a classroom setting.

#### 3.1.5. Step 5: Implementation

In the penultimate step of intervention mapping, a plan is created to structure the implementation of the intervention. The goal is for the intervention to be accepted and successfully integrated into educational routines to bring about long-term changes in perception. For example, in the designed intervention for destigmatizing palliative care, school administration should recognize its importance and ideally allow it to occur once per school year. If goals are achieved, the intervention should become a permanent part of the school curriculum. It is crucial to define who will deliver the intervention content. In schools and universities, trained teachers or specifically trained personnel from the intervention team itself could serve as interventionists. In total, three teams comprising three members each should be formed (one for schools, universities, and vocational institutions). Taking a closer look at schools, in Germany, there are usually five classes per year each comprising 25 students. Collaborating with three schools in the initial phases of the interventions seems feasible. Consequently, per year, each team should at least reach 375 young adults.

#### 3.1.6. Step 6: Evaluation Plan

Evaluation studies should be designed early in the planning process. The gold standard is randomized controlled trials, which are challenging to conduct in educational settings. Therefore, a mixed methods design is proposed. The systematic review identified the validated PaCKS questionnaire as suitable for assessing attitudes and knowledge about palliative care [44,45,46]. This is carried out in a pre-post design: the questionnaire is distributed and answered by participants before the intervention and again after the intervention. Participants are re-surveyed three and six months later to check the intervention’s long-term success. Increase in knowledge can further be tested through the development of a specific questionnaire that assesses knowledge using comparative self-assessment (CSA-gain) [47]. To add up, a change in attitude towards palliative care can be assessed using the validated CPD (“Continuing professional development”) Reaction Questionnaire [48,49]. This should be complemented by a qualitative study evaluating the perception of the intervention within focus group interviews [28,50].

The results of these evaluations will help further tailor the intervention and its components to the needs of young adults and provide insights for designing possible interventions for other target groups. Figure 4 summarizes the components of the developed intervention.

## 4. Discussion

To this day, palliative care faces stigmatization within the general public and young adults [41]. In the past, interventions to change the perception and increase knowledge of palliative care have been developed. As identified within the systematic review conducted by our working group, these interventions were mainly addressed towards the general public [41]. Moreover, evaluations demonstrating long-term results are lacking.

To promote the utilization of palliative care, broad programs aimed at destigmatization are necessary. However, in the context of public health, this proves challenging because interventions should target specific populations, such as young adults, yet the general population is very heterogeneous. It appears unrealistic that an intervention aimed at destigmatizing palliative care would reach all strata of the population effectively; therefore, the current focus should be on clearly defined target groups. Our approach to destigmatizing palliative care among the crucial peer group of young adults can be seen as novel.

Any intervention aiming at destigmatizing palliative care among young adults should incorporate multiple components. Merely imparting knowledge is unlikely to yield long-term success. Instead, a combination of various approaches appears to be more effective. Through the framework of Intervention Mapping, the following components have been developed in this work: educational training in institutions, creation of informational brochures, interactions with palliative care patients and personnel, public campaigns in communities via posters and social networks, and the establishment of a smartphone-friendly website with information.

Intervention Mapping is an appropriate model for designing a needs-based intervention to destigmatize palliative care among the peer group of young adults [31,32]. Environmental factors play a significant role in this context: peers, educators, and media were identified as crucial components that can support a long-term change in the perception of palliative care. To initiate this change, a suitable setting is required. Educational institutions provide a familiar and safe environment to discuss taboo topics [51]. Combining the dissemination of knowledge about palliative care within the protected framework of educational institutions, with the presence of the intervention through posters and brochures, as well as a public campaign, consolidates knowledge and encourages active engagement with palliative care content. This leads to discussions within the peer group, influencing attitudes and repeatedly bringing palliative care to mind from a young age. Key stakeholders include school directors and teachers, who also need to be trained beforehand to understand the topic’s relevance and possibly act as agents within the intervention. The shift to digital information acquisition is increasingly important, so print media should be supplemented with a web-based presence. New media, such as social networks (e.g., Instagram and TikTok), are preferred by young adults for receiving information and engaging in discussions with peers [52]. This enables the dissemination of knowledge to a broader audience. It has been shown that information on the internet is primarily accessed through devices available to the target population [35]. For young adults, these are predominantly smartphones, so an accompanying website must be optimized for mobile devices. Discussions about socially stigmatized topics like death and dying could be facilitated through a forum on this website. Additionally, information on counseling centers and palliative care addresses should be provided to ease access to palliative care.

As part of the intervention, young adults should directly interact with palliative care patients to reduce stigma around seriously ill individuals [28,32]. The systematic review identified fear associated with death, dying, and palliative care [41]. Interacting with palliative care patients who are benefiting from palliative care and maintaining quality of life despite their illness can allay these fears and destigmatize palliative care. To achieve this goal, key evidence-based change methods such as ‘interpersonal contact’ and ‘cooperative learning’ have been identified [31].

## 5. Limitations

Although all the results of the systematic review that informed this framework indicated that young people are less informed about palliative care than older populations, this was generally not specifically investigated as a primary research question. Therefore, the results of the needs assessment can only serve as a first approach. Future research on knowledge and perceptions of palliative care should focus specifically on young adults. Interventions with multiple components were rarely implemented. This makes it difficult to compare different interventions and their possible outcomes on a broad scale. The strategies developed within the framework of Intervention Mapping for creating a potentially suitable intervention and public campaign are based on concepts from public health research. However, they are only partially based on specific findings from research projects in the area of destigmatizing palliative care and can momentarily only serve as a first conceptualization of future campaigns.

## 6. Conclusions and Outlook

Even after more than 60 years of continuous development in palliative care, this young medical discipline is still burdened with multiple stigmas. Young people associate palliative care with death, dying, and end-of-life medical care, mainly for oncology patients. Palliative care is still often equated with hospice care. While young adults have often heard of palliative care, they lack detailed knowledge about its content [41]. Young adults receive little attention in interventions and public campaigns, making them poorly tailored to this target population.

It was the aim of this study to develop a theoretical approach aiming at destigmatizing palliative care among young adults. The key to any intervention and public campaign in the context of public health, aimed at achieving long-term changes in health behavior, is the accurate identification of the target population and its needs, desires, and expectations. The concept of Intervention Mapping provides a suitable framework for planning, implementing, and establishing interventions effectively. Participation of young adults in the planning of interventions and public campaigns is essential. If the proposed intervention was carried out, an early engagement with palliative care during school education could contribute to the long-term reduction of stigmas and address the demographic shift effectively. Resultantly, early access to palliative care promotes a self-determined and empowered life for young people coming of age. As a result of knowledge acquisition and a shifted perception of palliative care, the quality of life of young adults when facing severe illness improves. This has the potential to trigger reflected decisions about one’s own health, making health behavior a conscious process.

A multimodal intervention approach including knowledge dissemination, exchange, and media presence provides an appropriate framework to counter the existing stigmatization of palliative care within the peer group of young adults.

## Figures and Tables

**Figure 1 healthcare-12-01863-f001:**
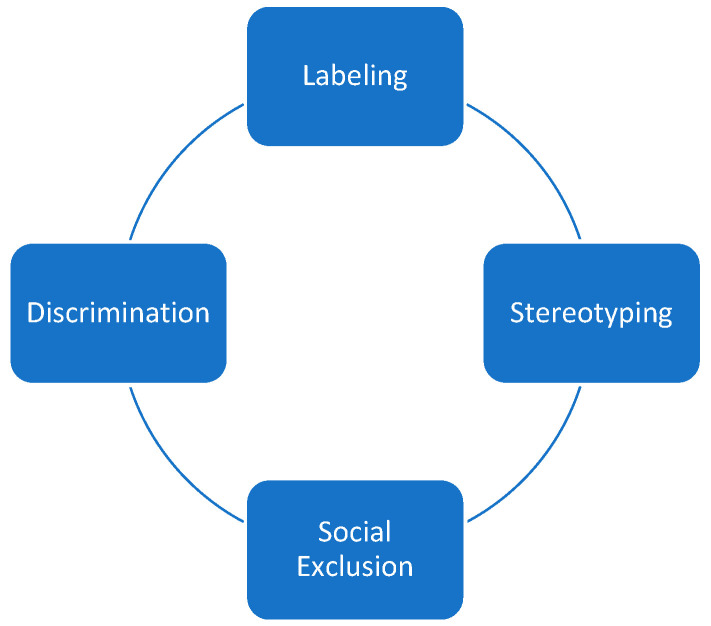
Link and Phelan’s conceptualization of stigmatization [12].

**Figure 2 healthcare-12-01863-f002:**
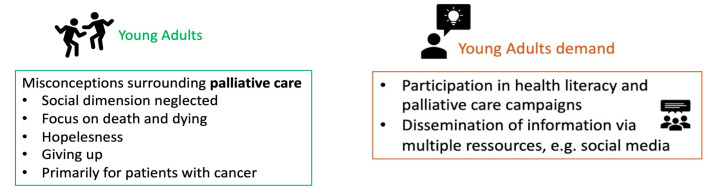
Young adults’ misconceptions of palliative care and their involvement in future campaigns [41].

**Figure 3 healthcare-12-01863-f003:**
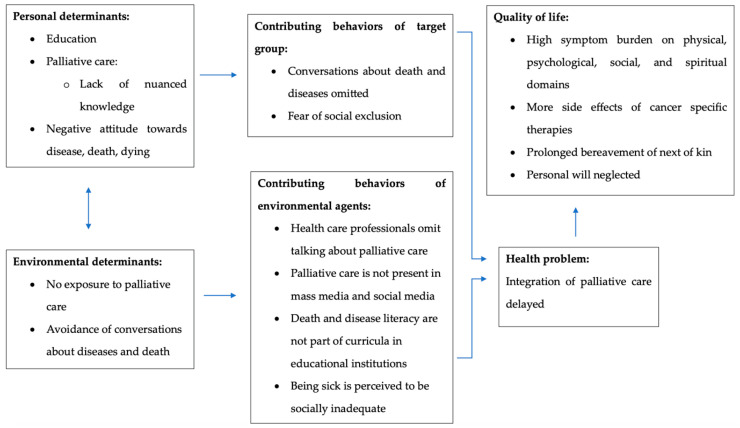
Logic model of the problem.

**Figure 4 healthcare-12-01863-f004:**
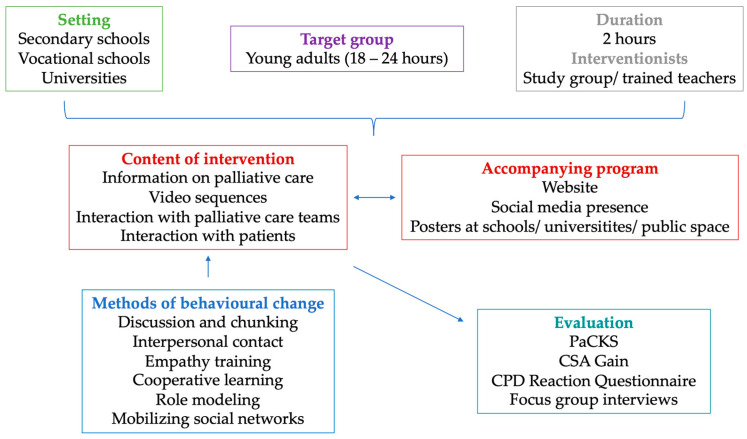
The developed theoretical framework for the intervention. (PaCKS = Palliative Care Knowledge Scale; CSA = comparative self-assessment; CPD = continuing professional development).

**Table 1 healthcare-12-01863-t001:** Planning matrix for changing personal determinants.

Performance Objective	Knowledge	Attitude	Values
Young adults talk about palliative medicine	Young adults recall core elements of palliative medicine in conversation.	Young adults express positive feelings regarding palliative care for improving quality of life.	Young adults recognize that palliative medicine is life-affirming.
Young adults talk about death and dying	Young adults know that there are symptom control options in the dying phase and fears are unfounded.	Young adults recognize that an open approach to death and dying leads to better end-of-life care.	Young adults recognize that death and dying are integral parts of life.

**Table 2 healthcare-12-01863-t002:** Planning matrix for changing environmental determinants.

Performance Objective	Knowledge	Attitude	Norms
Educational institutions enable training on palliative medicine	Detailed knowledge of palliative medicine is not known in the target population.	Palliative medicine is a relief for affected families. Young adults can focus on education.	Palliative medicine is a relevant medical discipline with equal importance as other disciplines.
Communities place posters in accessible locations	Communities are aware that knowledge about palliative medicine is only rudimentary in the general population.	Communities recognize that palliative medicine, like death and dying, is an integral part of the community.	Communities realize that palliative medicine should be sought earlier.
Interventionists distribute information on palliative care via the internet	Interventionists know about the specific needs of the target population.	Interventionists realize that the attitude of the target group must be altered.	Education on death, dying, and coping with diseases is as easily accessible as other content.

## Data Availability

The original contributions presented in the study are included in the article; further inquiries can be directed to the corresponding author.

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
