# Peer review of "Destigmatizing Palliative Care among Young Adults—A Theoretical Intervention Mapping Approach"

_healthcare, 2024, doi:10.3390/healthcare12181863_

Round 1

Reviewer 1 Report

Comments and Suggestions for Authors

As the authors of the manuscript correctly pointed out, the world is ageing rapidly and, simultaneously, the prevalence of oncological diseases is rising. For this reason palliative care has the potential to gain significance in many aspects of healthcare. These aspects are very comprehensively presented by the authors as well as why they are stigmatized. The authors offer a solution how to reach the population of young adults during school education through multimodal intervention approach to counter taboos around death and dying and to counter stigmatization of palliative care.

I have the following comments:

Line 64-65: „Treatment initiation is delayed are initiated belatedly, and stigmatized patients exhibit poorer compliance and therapy adherence due to feelings of shame” ” – this sentence is not fully comprehensible and needs correction.

Figure 4. Abbreviations should be defined.

„Empathy training” instead of „Empathy traning”

Line 437-439: „As identified within the systematic review conducted by our working group, these interventions were mainly addressed towards the general public or thorough evaluations are lacking” – this sentence is not fully comprehensible and needs correction.

Author Response

Thank you for your kind and constructive feedback. We were able to address all suggestions made and corrected the corresponding passages.

  • Line 64-65: „Treatment initiation is delayed are initiated belatedly, and stigmatized patients exhibit poorer compliance and therapy adherence due to feelings of shame” ” – this sentence is not fully comprehensible and needs correction.
    • The sentence was corrected.
    • Treatment initiation is delayed or initiated belatedly. Furthermore, due to feelings of shame, stigmatized patients exhibit poorer compliance and therapy adherence
  • Figure 4. Abbreviations should be defined.
    • The abbreviations were added.
  • „Empathy training” instead of „Empathy traning”
    • The term was corrected.
  • Line 437-439: „As identified within the systematic review conducted by our working group, these interventions were mainly addressed towards the general public or thorough evaluations are lacking” – this sentence is not fully comprehensible and needs correction.
    • The sentence was corrected.
    • As identified within the systematic review conducted by our working group, these interventions were mainly addressed towards the general public [41]. Moreover, evaluations demonstrating long-term results are lacking.

Reviewer 2 Report

Comments and Suggestions for Authors

General Comments:

This article addresses a number of gaps in the literature. The theoretical nature of the article, i.e., it outlines a process rather than research findings, is offset by the fact that the process may be revised and used in a variety if settings outside the occupational-cultural setting of the authors. This reviewer greatly appreciated the scope and detail of reflection that informs this article.

Specific points to consider:

2) I wonder if the authors can strengthen this paper with illustrations from their own use of this methodology.

Comments on the Quality of English Language

In general this paper is readable. At times the mode of presentation feels "thick." The authors may wish to check the reading level of their work.

I observed a number of sentences and paragraphs where the grammar or syntax felt awkward. The most significant example occurs in the Introduction and at least one "box" in the "Logic Model of the Problem." 

Please proofread for missing or extra words.

Author Response

Thank you for your feedback and taking the time to help improve our manuscript. The whole manuscript was revised accordingly.

  • I wonder if the authors can strengthen this paper with illustrations from their own use of this methodology.
    • We added this aspect.
    • At our faculty, this methodology is regularly used in the development of health-related interventions such as encouraging the general public at increasing daily steps. Currently, it serves as the basis for the development of a palliative care awareness month in Düsseldorf, Germany (work in progress).
  • In general this paper is readable. At times the mode of presentation feels "thick." The authors may wish to check the reading level of their work.
    • We revised the whole manuscript, edited sentences, left unnecessary words or sentences out to ensure a smoother reading experience.
  • I observed a number of sentences and paragraphs where the grammar or syntax felt awkward. The most significant example occurs in the Introduction and at least one "box" in the "Logic Model of the Problem."
    • The “logic Model of the Problem” was edited, the introduction and other parts of the manuscript were revised for better readability. Grammar and syntax have both been checked and were adapted accordingly.
  • Please proofread for missing or extra words.
    • We proofread the article.

Reviewer 3 Report

Comments and Suggestions for Authors

-          I sincerely thank the authors of the paper for their contribution on one of the great challenges in public health nowadays: the integration of PC and EOL care. The article is well written, it is robust and is supported by a previous systematic review on the topic. It provide a correct insight into the theoretical framework adopted (Intervention Mapping approach). Here some thoughts and simply comments about the manuscript.

-          Even if author mainly focus on the theoretical aspects, I could find many references to the methodology that would be adopted in implementing it. I wonder if author could provide some more details on this aspect: number of people targeted, numbers about the teams involved, an estimation of the costs of such interventions, the techniques authors would apply to promote interactions between young adults and PC patients (which author identify as one of the way to de-stigmatize PC interventions).

-          I partially disagree with the stringent idea that death and dying are "suppressed from daily life". I think the issue is more complex. For example, with the COVID-19 pandemic, in many countries death and dying that have resurfaced forcefully in daily life. Narratives about wars and terrorist attacks are often spread by social media and other mass media. Illnesses, cancer patients, and hospice are more often part of our environment and PC services are seldom taking care of the parents and grandparents of the young adults. At the same time, an aspect much less considered is the way our societies organize the death and dying processes, the way our society talk about those social facts and, moreover, the way our society and their members take care (and are able to) of the more fragile members of society itself.

-          Consequently, from previous comment, since the aim of the intervention is to tackle the stigma of early integration of PC within care pathways, I wonder if another important leverage weren’t professionals, especially GPs, nursing homes’ personnel, and those who operate at community level, which generally should refer patients to PC services or specialists. Is the project tackling with these aspects? Are the PC services ready to introduce PC earlier in the care pathways? Are those services ready to take care of non-cancer patients, which would dramatically increase the number of patients (and families) which every year would be in need of PC?

-          Considering the topic, I would suggest the authors to refer to Kellehear public health approach on palliative care as a one of the cornerstones on this matter.

Author Response

Dear reviewer,

Thank you for your time and the effort undertaken to help improve our work. We were able to include most of the suggestions made and feel our manuscript gained significantly in quality. Thank you for that.

  • Even if author mainly focus on the theoretical aspects, I could find many references to the methodology that would be adopted in implementing it. I wonder if author could provide some more details on this aspect: number of people targeted, numbers about the teams involved, an estimation of the costs of such interventions, the techniques authors would apply to promote interactions between young adults and PC patients (which author identify as one of the way to de-stigmatize PC interventions).
    • We were able to address all aspects but the aspect of costs since, from faculty to faculty in Germany, different costs are allocated. We were very happy to include the other aspects you mentioned as we feel this helps transposing the theory into practice:
      • Patients with stable diseases and overall well-being could be present at educational in-stitutions during the course of the intervention. They can be recruited through palliative care teams serving as advisors.
      • In total, three teams comprising of three members each should be formed (one for schools, universities, and vocational institutions). Taking a closer look at schools, in Germany, there are usually five classes per year each comprising 25 students. Collaborating with three schools in the initial phases of the interventions seems feasible. Consequently, per year, each team should at least reach 375 young adults.
    • I partially disagree with the stringent idea that death and dying are "suppressed from daily life". I think the issue is more complex. For example, with the COVID-19 pandemic, in many countries death and dying that have resurfaced forcefully in daily life. Narratives about wars and terrorist attacks are often spread by social media and other mass media.
      • We changed the sentence and mentioned the ongoing shift in the presence of death in everyday life through media:
        • Due to the Covid pandemic and a rising prevalence of international conflicts, diseases and death are more present in the mass media and, as a result, in people’s lives. However, in personal spaces, death and dying have been largely are still mainly avoided in everyday life.
      • Illnesses, cancer patients, and hospice are more often part of our environment and PC services are seldom taking care of the parents and grandparents of the young adults. At the same time, an aspect much less considered is the way our societies organize the death and dying processes, the way our society talk about those social facts and, moreover, the way our society and their members take care (and are able to) of the more fragile members of society itself. Consequently, from previous comment, since the aim of the intervention is to tackle the stigma of early integration of PC within care pathways, I wonder if another important leverage weren’t professionals, especially GPs, nursing homes’ personnel, and those who operate at community level, which generally should refer patients to PC services or specialists. Is the project tackling with these aspects? Are the PC services ready to introduce PC earlier in the care pathways? Are those services ready to take care of non-cancer patients, which would dramatically increase the number of patients (and families) which every year would be in need of PC?
        • All the named aspects are of great relevance in public health. We agree that they need further addressing in the future, however, our work focuses on the general public and young adults. For example, when conducting the systematic review this work is based on, we consciously removed health care professionals from the developed search string. It is very interesting to develop a campaign that addresses health care personnel and then link those results to the results in this manuscript. We feel like this is a starting point and the basis for future work. Thank you for the inspiration.
      • Considering the topic, I would suggest the authors to refer to Kellehear public health approach on palliative care as a one of the cornerstones on this matter.
        • Thanks for suggesting to include Kellehear’s public health approach. We found an interesting review that we cited:
          • It is in accordance to Kellehear’s ‘new public health’ approach to palliative care, as affected individuals seek support from peers, family, and friends in severe disease tra-jectories [43]. To empower them and further develop crucial networks, so called compassionate communities are developed. It is them who need to play an active role in the development of a novel intervention.

Reviewer 4 Report

Comments and Suggestions for Authors

Dear authors:

Congratulations on your work.

Here are some comments/questions:

Abstract is too long. Only 250 words are allowed.

Why did you choose to target individuals from 18 to 24 years old?

Methodology needs further explanation. You state you did a systematic review was done. How was it done? what methodology was used?

No conclusions are presented. In the outlook section, you do not respond to your main aim. What are the implications for practice and population if this campaign was done?

Comments on the Quality of English Language

 Minor editing of English language required.

Author Response

Thank you for your kind and constructive feedback. We were able to address all suggestions made and corrected the corresponding passages.

  • Abstract is too long. Only 250 words are allowed.
    • We shortened the abstract to 300 words. On Healthcare’s website it says that “the abstract should be a total of about 250 words” so we hope that this number of words is still acceptable.
  • Why did you choose to target individuals from 18 to 24 years old?
    • We chose this target population since we feel people in this age group can still be effectively reached through schools and universities. Moreover, their health behavior is still influenceable.
      • The German Federal Ministry of Justice and Consumer Protection, for instance, de-fines individuals aged 18 to 27 as "young people" [42]. Internationally, there is no uniform definition, so for this study, the age range was adjusted: individuals aged 18 to 24 years were defined as the target population as they can be systematically reached in educational and vocational institutions.
    • Methodology needs further explanation. You state you did a systematic review was done. How was it done? what methodology was used?
      • We further explained this.
        • On the basis of the PRISMA Checklist and according to the PICOS process, a search string was developed. Within Medline (via Pubmed), google scholar, and Web of Science, relevant literature addressing young adults’ knowledge and perception of palliative care was screened by two reviewers.
      • No conclusions are presented. In the outlook section, you do not respond to your main aim. What are the implications for practice and population if this campaign was done?
        • The outlook was adapted accordingly.
          • It was the aim of this study to develop a theoretical approach aiming at destigmatizing palliative care among young adults […] If the proposed intervention was carried out, an early engagement with palliative care during school education could contribute to the long-term reduction of stigmas and address the demographic shift effectively. Resultantly, early access to palliative care promotes a self-determined and empowered life for young people coming of age. As a result of knowledge acquisition and a shifted perception of palliative care, the quality of life of young adults when facing severe illness improves. This has the potential to trigger reflected decisions about one’s own health, making health behavior a conscious process.

Round 2

Reviewer 4 Report

Comments and Suggestions for Authors

Dear authors:

Congrats on your work.

It was improved since the last review.

Best regards.

Comments on the Quality of English Language

Minor editing of English language required.

There are also some errors/typos on the writting.